# Ankrd1 Promotes Lamellipodia Formation and Cell Motility via Interaction with Talin-1 in Clear Cell Renal Cell Carcinoma

**DOI:** 10.3390/ijms26094232

**Published:** 2025-04-29

**Authors:** Yuki Takai, Sei Naito, Hiromi Ito, Shigemitsu Horie, Masaki Ushijima, Takafumi Narisawa, Mayu Yagi, Osamu Ichiyanagi, Norihiko Tsuchiya

**Affiliations:** Department of Urology, Faculty of Medicine, Yamagata University, 2-2-2 Iida-nishi, Yamagata 990-9585, Japan; seinaitoh@yahoo.co.jp (S.N.); ito.hiromi@med.id.yamagata-u.ac.jp (H.I.); hakuro19811113@yahoo.co.jp (S.H.); uroushi@gmail.com (M.U.); tnari_0623@yahoo.co.jp (T.N.); mayu11ism05@gmail.com (M.Y.); oichiyan@ab.cyberhome.ne.jp (O.I.); norihiko.tsuchiya@gmail.com (N.T.)

**Keywords:** renal cell carcinoma, metastasis, lamellipodia

## Abstract

Ankyrin repeat domain 1 (Ankrd1), a transcriptional target of Yes-associated protein (YAP), is linked to cardiomyopathy. However, its role in cancer, particularly in clear cell renal cell carcinoma (ccRCC), remains vague. In this study, we examined the expression, regulation, and function of Ankrd1 in ccRCC. High Ankrd1 expression was related to poor prognosis in patients with ccRCC in The Cancer Genome Atlas cohort. Ankrd1 expression was regulated by YAP in all ccRCC cell lines examined and also by ERK5 in a subset of ccRCC cell lines. Moreover, silencing of Ankrd1 in ccRCC cell lines resulted in decreased cell motility, whereas its overexpression increased the cell motility. Ankrd1 colocalized with F-actin in lamellipodia upon phorbol ester stimulation. Ankrd1 silencing resulted in alterations in the shape of RCC cells and caused a decrease in lamellipodia formation. Ankrd1 also colocalized with talin-1 in lamellipodia. Ankrd1 depletion repressed talin-1-mediated activation of the integrin pathway. Immunohistochemical examination of surgical specimens revealed high expression of Ankrd1 in metastatic RCC tissues compared with that in primary RCC tissues from the same patients. Collectively, these findings suggest that Ankrd1 plays a critical role in the motility of ccRCC cells through lamellipodia formation.

## 1. Introduction

Epithelial–mesenchymal transition (EMT), characterized by depolarization of epithelial cells, loss of cell–cell adhesions, and transformation into fibroblast–like morphology, was previously believed to be essential for cancer cell metastasis. Cadherin switching—downregulation of E-cadherin and upregulation of N-cadherin—was considered a hallmark of EMT. However, recent reports suggest that EMT is not essential for metastasis [1,2]. N-cadherin is the predominant cadherin in clear cell renal cell carcinoma (ccRCC), being highly expressed in renal proximal tubular cells in which this carcinoma originates [3,4]. RCC exhibits strong mesenchymal features in comparison with other epithelial tumors, perhaps reflecting the fact that the kidney epithelium derives from condensation of mesodermal mesenchymal cells [5]. Moreover, the mesenchymal features do not relate to poor prognosis in patients with RCC.

The movement of cancer cells and the formation of metastatic lesions is a multistep process [6]. In mesenchymal motility, membrane protrusions, such as lamellipodia, filopodia, and invadopodia, are formed by actin polymerization [7,8]. Lamellipodia are sheet-like protrusions composed of a crisscross arrangement of actin filaments. Formation of lamellipodia at the leading edge of cells is the initial step in crawling cell motility. Filopodia are finger-like protrusions composed of parallel bundles of actin filaments that play roles in pathfinding and guidance towards chemoattractants. Invadopodia are protrusions of the basal cell surface and mediate degradation of extracellular matrix (ECM), such as the basement membrane. Integrin-dependent focal adhesions are necessary to attach these protrusions to the ECM [6]. Focal adhesions connect the actin cytoskeleton of membrane protrusions to the ECM, stabilizing the protrusions. Thereafter, retraction of the rear edge of cells to follow the leading edge provides forward propulsion. Mesenchymal cells can move through the ECM by degrading it via proteolysis. In addition to the motility of individual cells, several other factors, such as infiltration to lymphovascular vessels, attachment to vascular endothelial cells, and survival in distant organs, are involved in metastasis. In RCC metastasis, loss of intercellular attachment represented by the cadherin switching, might not be important, and any of the other processes mentioned above might perform key functions in RCC cell motility and metastasis.

Ankyrin repeat domain 1 (Ankrd1) is a cardiac-specific stress-response protein that plays critical roles in transcriptional regulation, myofibrillar assembly, stretch sensing, and communication between the sarcoplasm and nucleus during heart development and cardiac insults [9]. Ankrd1 has protein–protein interacting domains, including four ankyrin repeats and a coiled-coil domain. Ankrd1 can interact with transcription factors (YB-1), myofibrillar proteins (titin and myopalladin), intermediate filaments (desmin), calcium-handling proteins (cardiac calsequestrin 2), and ubiquitin ligases (muscle-specific RiNG finger 1 and 2). Ankrd1 does not have kinase activity; however, it acts as an adaptor protein by interacting with the abovementioned proteins [9]. Ankrd1 shuttles between the cytoplasm and nucleus as part of a stress-related regulatory pathway in the myocardium [10]. In the cytoplasm, Ankrd1 interacts with titin and myopalladin to maintain sarcomere organization [11]. Ankrd1 is a transcriptional target of YAP, a major effector of the Hippo pathway [12]. We previously reported that Ankrd1 is induced by ERK5 in rat pheochromocytoma (PC12) cells [13] and could detect Ankrd1 expression in surgical specimens of composite pheochromocytoma and ganglioneuroblastoma [14]. However, there are no reports on any other expression regulation mechanism.

Ankrd1 is associated with the pathogenesis of cardiomyopathy [11]. However, its role in cancer has scarcely been investigated. Ankrd1 was found to be associated with tumor progression and drug resistance in some studies. Overexpression of Ankrd1 was related to EMT promotion in lung and breast cancers [15,16]. The expression and function of Ankrd1 have not been investigated in RCC. In the present study, we examined the expression and function of Ankrd1 in ccRCC, especially the effects of this protein on cell motility and metastasis, and deciphered the regulatory mechanisms involved.

## 2. Results

### 2.1. Prognostic Relevance of Ankrd1 Expression in TCGA-ccRCC Data Sets

We explored the existence of any relationship between *Ankrd1* expression and prognosis of RCC using data downloaded from The Cancer Genome Atlas (TCGA) for patients with ccRCC. Based on the Kaplan–Meier survival analysis, ccRCC patients with high *Ankrd1* expression had an unfavorable overall survival rate (log-rank *p* = 0.012; Appendix A). The results indicated that Ankrd1 is associated with the progression of ccRCC, as in the case of other cancers.

### 2.2. Expression of Ankrd1 Is Regulated by YAP1 and ERK5

We found that Ankrd1 is expressed in RCC cell lines ACHN, 769-P, 786-O, A-498, Caki-1, and Caki2 (Figure 1A). To characterize the function of Ankrd1 in RCC cell lines, we use two representative Ankrd1-expressing models of ccRCC, namely 786-O and 769-P. The A-498 line, with low levels of endogenous Ankrd1, was used to evaluate the effects of Ankrd1 overexpression.

We investigated the viability of 786-O and 769-P cells transfected with siRNA against Ankrd1. RT-qPCR and Western blotting analyses showed significant inhibition of Ankrd1 expression by siAnkrd1 (Figure 1B). When the cells were seeded at high density (3 × 10^3^ cells/well), Ankrd1 depletion did not cause changes in cell viability (Figure 1C, left). However, depletion of Ankrd1 suppressed the viability of 786-O cells seeded at low density (0.8 × 10^3^ cells/well) (Figure 1C, right). Ankrd1 is a transcriptional target of YAP, a major effector of the Hippo pathway [12]. At high cell density, activated LATS1/2 kinases phosphorylate YAP. Phosphorylation of YAP^Ser127^ leads to sequestration of YAP in the cytoplasm and subsequent inhibition of transcription of its target genes. When the confluency of 786-O, 769-P, and A-498 cells was increased from 50% to 100%, total YAP levels were not changed; however, phosphorylation of YAP^Ser127^ was increased (Figure 1D). Moreover, in 786-O and 769-P cells, *Ankrd1* mRNA expression was decreased by half when cells reached confluency (Figure 1E). Phosphorylation leads to cytoplasmic translocation and inactivation of YAP, which reduces Ankrd1 expression. At low cell density, the expression of Ankrd1 is high because YAP is activated (non-phosphorylated). Therefore, knockdown of Ankrd1 significantly affected cell viability at low cell density when Ankrd1 expression was high. *Ankrd1* mRNA levels in 786-O and 769-P cells decreased significantly 72 h after treatment with YAP inhibitors, verteporfin and K975 (Figure 1F). Verteporfin and K975 inhibit the transcription of YAP target genes by blocking the binding of YAP to TEAD factors that act as its coactivator. These results indicated that YAP regulates *Ankrd1* expression in ccRCC, as it does in several other cancer cells.

We previously reported that Ankrd1 is induced by ERK5 in rat pheochromocytoma (PC12) cells [13]. Our laboratory also recently reported that strong ERK5 expression in surgical RCC specimens was associated with high recurrence rates, and that ERK5 inhibition decreased cell proliferation and survival in ccRCC cells [17]. In this context, we investigated the effects of inhibiting ERK5 on Ankrd1 expression in RCC cell lines. 786-O and 769-P cells were exposed to two ERK5 inhibitors, XMD8-92 and XMD17-109, for 48 h (Figure 1G). Optimal doses of XMD8-92 and XMD17-109 were set based on their cytotoxicity, as assessed using the MTS assay (Appendix A). Both the inhibitors suppressed *Ankrd1* expression in 786-O cells. In 769-P, *Ankrd1* expression was slightly reduced by XMD17-109 but not by XMD8-92, which indicated that at least in 786-O, Ankrd1 expression is regulated not only via the Hippo pathway but also via another pathway specific to cancer cells.

In myoblasts (C2C12 cells), YAP promotes cell differentiation by activation of the MEK5-ERK5 pathway [18]. Ippolito et al. demonstrated that ERK5 modulates YAP activity in a Hippo-independent manner in hepatocytes [19]. In 786-O and 769-P cells, expression of ERK5 and pERK5 was not changed by verteporfin treatment (Figure 1H). Also, XMD8-92 and XMD17-109 did not change the phosphorylation status of YAP (Figure 1I). These results indicate that ERK5 and YAP independently regulate Ankrd1 expression.

### 2.3. Knockdown of Ankrd1 Decreases Cell Proliferation

We observed an inconsistent change in the expression of apoptotic markers upon Ankrd1 silencing (Figure 2A). Among the antiapoptotic markers, the expression of Mcl-1 was significantly decreased upon Ankrd1 silencing, whereas that of Bcl-2 was slightly increased. Cleavage of caspase-3 and PARP was not observed. Thus, we did not find any evidence of Ankrd1 inducing apoptosis in RCC cell lines. We evaluated the effects of Ankrd1 on the cell cycle using flow cytometry. Ankrd1 silencing increased the proportion of G0/G1 cells and reduced the proportion of cells in the S phase at 72 h, albeit nonsignificant (Figure 2B). Seeing that Ankrd1 knockdown decreased the viability of 786-O and 769-P cells only at a low cell density, apparently Ankrd1 has a small impact on the viability of RCC cells.

### 2.4. Ankrd1 Modulates the Migration and Invasion of RCC Cells

Next, we examined the role of Ankrd1 in the motility of RCC cells. Ankrd1 knockdown significantly reduced the migration ability of 786-O and 769-P cells in the wound-healing assay (Figure 3A). Consistent with this, overexpression of Ankrd1 remarkably increased the migration of A-498 cells (Figure 3B). In the invasion assay, knockdown of Ankrd1 reduced the invasive ability of 786-O and 769-P cells, whereas its overexpression increased the invasive ability of A-498 cells (Figure 3C,D). These results indicate that Ankrd1 plays a role in the migration and invasion of RCC cells.

We investigated the link between Ankrd1 expression and EMT-related proteins. Expression of N-cadherin, E-cadherin, and vimentin was not changed by knockdown or overexpression of Ankrd1 (Figure 3E). ccRCC cells are known to poorly express E-cadherin [20]. In this study as well, E-cadherin expression was not observed regardless of the knockdown or overexpression of Ankrd1. Based on these results, we believe that Ankrd1 participates in RCC metastasis without the involvement of the cadherin switch.

### 2.5. Interaction of Ankrd1 and Talin-1 Is Involved in the Formation of Lamellipodia

We noted that siAnkrd1 transfection altered the morphology of 786-O cells (Figure 4A). The cells became larger and appeared fusiform. We hypothesized that Ankrd1 participates in the maintenance of cell morphology and motility via the regulation of cytoskeleton remodeling.

12-O-Tetradecanoylphorbol 13-acetate (TPA) stimulation triggered the formation of membrane protrusion in 786-O cells, and Ankrd1 was found to be localized in these protrusions (Figure 4B). TPA, the most commonly used phorbol ester, is a PKC activator. PKC induces cell motility via activation of Src, which enhances the formation of lamellipodia and invadopodia, as well as membrane ruffling [21,22,23]. Untreated 786-O cells showed typical actin stress fibers. After 20 min of TPA exposure, strong actin staining was evident in membrane protrusions, and stress fibers were significantly reduced. TPA-induced membrane protrusions, identified by colocalization of F-actin and cortactin arrangement, were considered to be lamellipodia. In addition, some cells displayed individual invadopodia and invadopodial rosettes (Figure 4C). Pretreatment of 786-O cells with siAnkrd1 or the YAP inhibitor, verteporfin, blocked TPA-induced lamellipodia formation (Figure 4C,D). These data indicate a potential role for Ankrd1 in the formation of lamellipodia and actin remodeling.

Ankrd1 contains tandem ankyrin repeats and a coiled-coil domain. These domains mediate protein–protein interactions, and thereby Ankrd1 can interact with a large variety of proteins. In cardiac muscle cells, Ankrd1 interacts with the cytoskeleton via its binding with sarcomeric proteins to maintain sarcomere organization. In cardiac muscle cells, Ankrd1 is also found to bind to talin-1 [24]. Talin-1 mediates cell–ECM adhesion by linking integrins to the actin cytoskeleton. In this study, Ankrd1 and talin-1 colocalized in the TPA-induced lamellipodia of 786-O cells (Figure 4E).

Using anti-Flag-mediated immunoprecipitation of proteins in 786-O cells transfected with Flag-Ankrd1, we found that Ankrd1 is isolated in a complex with talin-1 and β-actin (Figure 4F). Consistent with the coimmunoprecipitation results, we observed positive PLA signals with Ankrd1 and talin-1 antibodies in 786-O cells (Figure 4G).

In several cancer cell lines, talin-1 is necessary for hyaluronic acid (HA)-induced activation of integrin [25,26]. We determined the phosphorylation status of the known substrates of HA-induced integrin pathways, non-receptor tyrosine kinases Src and FAK (Figure 4H). The phosphorylation of Src at the activation loop of its kinase domain (Tyr 416) and of FAK at the autophosphorylation site (Tyr397) reached a peak at 45 min after HA stimulation. Silencing of Ankrd1 attenuated the phosphorylation of Src and FAK. Therefore, Ankrd1 might be indirectly involved in activation of the integrin pathway via interaction with talin-1.

Talin-1 triggers the recruitment of vinculin, another actin-binding protein, to focal adhesion (FA). Formation of integrin-talin-1-vinculin complex strengthens the connection of integrin with actin and matures adhesion [27]. We evaluated the change in localization of vinculin during cell migration using immunofluorescence of vinculin and phalloidin (Figure 4I). In 786-O cells, vinculin was localized to the cell perimeter and also to the tips of stress fibers, which is the typical site for FA distribution. After TPA treatment, vinculin was localized to the leading edge, similar to phalloidin. Knockdown of Ankrd1 attenuated the change in localization of vinculin by TPA.

### 2.6. Ankrd1 Is Highly Expressed in Metastatic RCC Tissues

To determine whether Ankrd1 participates in RCC metastasis, expression of Ankrd1 in metastatic RCC lesions was assessed using immunohistochemistry (IHC). Patient demographics are summarized in Table 1. Representative images of IHC staining are shown in Figure 5A. In normal renal tissue, Ankrd1 was expressed in renal tubular epithelial cells but not in glomerulus and interstitial tissue. RCC tissues showed Ankrd1 staining in the membrane and nucleus of RCC cells. Considering the function of Ankrd1 in the cytoplasm as indicated in in vitro experiments, we focused on the membrane-associated Ankrd1.

Expression of Ankrd1 in the RCC cell membrane was assessed using the IHC scores. Representative IHC images used to grade Ankrd1 expression in ccRCC tissues are shown in Figure 5B. We compared the IHC scores of metastatic and primary RCC lesions of identical patients (Figure 5C). The average IHC score was significantly higher for metastatic lesions (8.65 for metastatic lesions vs. 4.40 for primary lesions, *p* < 0.01). Almost all patient pair samples displayed higher expression of Ankrd1 in the metastatic lesion compared with that in the primary lesion, except for three patients with lung, pancreas, and adrenal gland metastases. Ankrd1 expression in RCC tissues displayed intratumor heterogeneity. In particular, in the primary tumors, clear intratumor heterogeneity was observed in many more samples (14/20 (70.0%) of primary lesions vs. 5/26 (19.2%) of metastatic lesions).

We further evaluated Ankrd1 expression in primary RCC samples by adding samples from patients without metastases. The IHC score was not related to the Fuhrman grade and pT stage (Table 2). Thus, Ankrd1 might participate in metastasis independent of pathological malignancy. We compared the IHC scores of primary tumors from patients with and without metastasis. The average IHC score was equivalent in both groups (4.40 in patients with metastasis vs. 4.01 in patients without metastasis, *p* = 0.525).

These results indicate that Ankrd1 expresses heterogeneously in RCC tissues, and some cell groups with high expression might contribute to metastasis.

## 3. Discussion

We demonstrate that Ankrd1 expression in RCC cells is regulated by YAP and, depending on the cell line, by ERK5. We also found that Ankrd1 forms a complex with F-actin and talin-1 and promotes cell motility via the formation of lamellipodia. Moreover, the finding that the metastatic RCC lesions express high levels of Ankrd1 compared with the primary RCC samples from the same patients indicates that Ankrd1 participates in metastasis. To the best of our knowledge, this is the first study to explore the function of Ankrd1 in RCC cells and to show that Ankrd1 plays an essential role in cancer cell invasion via lamellipodia formation.

We found that Ankrd1 overexpression induced, and its silencing reduced, the migration and invasion of RCC cells. In addition, Ankrd1 interacted with actin, and Ankrd1 silencing inhibited the formation of TPA-induced lamellipodia, which involved actin polymerization. In a similar finding, Ankrd1 promoted cell motility via actin remodeling in fibroblasts during the wound-healing process [28]. Ankrd1-knockout mice exhibited a delayed-wound-closure phenotype. Skin fibroblasts with Ankrd1 deletion did not migrate or contract as efficiently as fibroblasts with Ankrd1 expression. In contrast, Ankrd1 overexpression induced the formation of an abundant actin network and contraction in fibroblasts.

We found that Ankrd1 also forms a complex with talin-1 in RCC cells. Ankrd1 binds to talin-1 in cardiac muscle cells, and *Ankrd1* mutations in dilated cardiomyopathy result in a loss of this binding [24]. Talin-1 and Ankrd1 play a role as a cellular mechano-sensory unit [11]. Talin-1 is a cytoplasmic adaptor protein that interacts with integrin β1, which is an adhesion molecule related to FA [27]. Talin-1 binds to the cytoplasmic tail of integrin β1 and facilitates its activation and linking to cytoskeletal actin. Talin-1 regulates the formation of FA via the recruitment of vinculin and microtubules to adhesion sites. Talin-1 promotes metastasis in several cancers [29,30,31,32]. In ccRCC, high expression of talin-1 is associated with microvascular and Gerota’s fascia invasion and poor prognosis [33]. At the leading edge of a cell, the integrin-talin-1-actin complex forms the core of FA, and then talin-1 recruits vinculin to the FA [34,35]. Activated vinculin binds to the Arp2/3 complex, which stimulates actin polymerization. The elongation of F-actin in the FA is related to membrane protrusion at the leading edge. Thus, talin-1 induces cell migration via regulation of the adhesion dynamics. In breast cancer, knockdown of talin-1 and blockade of its binding with integrin interfered with the formation of FA and the FAK-AKT signaling pathway [29]. This study revealed that knockdown of Ankrd1 suppressed the accumulation of vinculin at the leading edge and HA-induced phosphorylation of Src and FAK, the downstream effectors of the integrin pathway. These results indicate that Ankrd1 indirectly regulates the integrin pathway via its interplay with talin-1.

We found that Ankrd1 silencing inhibits the motility of RCC cells but did not alter the expression of EMT-related molecules. In previous reports on other cancers, Ankrd1 was found to be associated with EMT. In lung cancer, cell lines with high Ankrd1 expression exhibited higher migration abilities, and Ankrd1 expression was regulated by the EMT marker ZEB1 [12]. In breast cancer, Ankrd1 expression is positively regulated by Snail, which induces EMT [13]. RCC and proximal tubule cells, from where RCC originates, exhibit mesenchymal features. N-cadherin is highly expressed in normal renal proximal tubule cells. However, E-cadherin, which is expressed in other nephron segments, including the distal tubule, is rarely expressed in proximal tubule cells [3]. Compared with E-cadherin-containing adherens junctions, loose intercellular adhesion because of N-cadherin can facilitate transcellular absorption at proximal tubules [36]. Therefore, cadherin switching, which is a hallmark of EMT, is not essential for the acquisition of metastatic ability in RCC cells. Moreover, the mechanism underlying the promotion of RCC cell motility by Ankrd1 is different from the loss of intercellular adhesion as observed in cadherin switching.

Although Ankrd1 was expressed in normal renal tubular epithelial cells, as evident from IHC results, there is no report regarding the physiological functions of Ankrd1 in renal tubular cells. Moreover, Ankrd1 expression is regulated by YAP in RCC cells, similar to that in other kinds of cells. Although YAP mutations have not been reported in RCC, there are a few reports about the alteration of the regulatory mechanism of YAP. Merlin (*Neurofibromin2*, *NF2*) is involved in the inactivation of YAP via activation of the Hippo pathway. NF2 mutations were found in ccRCC [37]. YAP is also regulated by mechanical stimulations caused by changes in cell–cell or cell–ECM adhesion [14]. YAP responds to changes in ECM stiffness. A rigid ECM because of inflammation and deposition of collagen ensures nuclear localization of active YAP. Interaction of YAP with the integrin pathway has also been reported. Adhesion of integrin to fibronectin activates FAK/Src signaling. Activated FAK inhibits YAP phosphorylation and leads to the nuclear localization of YAP [38]. In this study, silencing of Ankrd1 inhibited Src and FAK phosphorylation. This result implied an interaction between the integrin and YAP pathways, as reported previously. Expression of Ankrd1 was also regulated by ERK5, independent of YAP in 786-O cells. However, in 769-P cells, regulation of Ankrd1 by ERK5 was not significant. A-498 cells expressed low levels of Ankrd1 compared with 786-O and 769-P. These results suggest that the system for regulating Ankrd1 expression differs among cell lines. We need to investigate the interaction of Ankrd1 expression with some other pathways enhanced in RCC.

We found that the impact of Ankrd1 on RCC cell proliferation might be small because Ankrd1 silencing suppressed the viability of cells only at low cell density. Additionally, Ankrd1 was not found to be involved in apoptosis. The function of Ankrd1 in apoptosis is controversial. Ankrd1 induces apoptosis in ovarian and non-small cell lung cancer [15,39,40]. However, Ankrd1 has also been reported to function as a tumor suppressor. Epigenetic inhibition of Ankrd1 correlates with the progression of pancreatic cancer [41]. In IHC evaluation, Ankrd1 expression was significantly higher in metastatic lesions compared with that in primary lesions. However, no correlation between Ankrd1 expression and grade or T stage was noted. These results are consistent with the in vitro data that Ankrd1 promoted cell motility but had little effect on cell proliferation. The results of Kaplan–Meier survival analysis indicated that high *Ankrd1* expression in ccRCC from the TCGA cohort is associated with poor outcome. However, in our cohort, Ankrd1 expression in primary RCC tissues was equivalent among patient groups with or without metastasis. It might be possible that the difference between the groups was masked because the patients with limited follow-up periods might experience recurrence in the future. Moreover, the expression of Ankrd1 was dynamically regulated by YAP and ERK5 in RCC cell lines and exhibited intratumor heterogeneity in ccRCC tissues. These factors made it hard to quantitatively evaluate the expression of Ankrd1 in RCC cell lines and RCC tissues.

This study had a limitation. It was noted that Ankrd1 indirectly regulates the integrin pathway via talin-1. However, the data are insufficient to verify its direct interaction with integrin and the subsequent actin dynamics, warranting further investigation.

## 4. Materials and Methods

### 4.1. Prognostic Analysis Using the Cancer Genome Atlas Database

We collected clinical data for ccRCC from TCGA (https://cancergenome.nih.gov/, accessed on 12 October 2022) using the cBioPortal (http://www.cbioportal.org/, accessed on 12 October 2022). The mRNA data for Ankrd1 for the corresponding patients (*n* = 510) were also downloaded from TCGA. We divided the patients into two groups according to the *Ankrd1* mRNA levels (high or low) and then compared the survival risk based on *Ankrd1* expression using the Kaplan–Meier analysis.

### 4.2. Cell Culture and Reagents

The RCC cell lines ACHN (RRID:CVCL_1067), 786-O (RRID:CVCL_1051), 769-P (RRID:CVCL_1050), A-498 (RRID:CVCL_1056), Caki-1 (RRID:CVCL_0234), and Caki-2 (RRID:CVCL_0235) were obtained from the American Type Culture Collection (Manassas, VA, USA). The cells were cultured as described previously [42]. In most experiments, 786-O, 769-P, and A-498 cell lines were used as ccRCC models [20]. We used two ERK5 inhibitors, XMD8-92 (ChemScene LLC, Monmouth Junction, NJ, USA; CS-0245) and XMD17-109 (Sigma-Aldrich, Merck, Darmstadt, Germany; SML1753), and two Yes–associated protein (YAP) inhibitors, verteporfin (ChemScene LLC; CS-1950) and K975 (MedChemExpress, Monmouth Junction, NJ, USA; HY-138565). TPA (catalog no. 4174) was purchased from Cell Signaling Technology Japan, Tokyo, Japan. HA (mol wt 15,000–30,000; catalog no. 97616) was obtained from Sigma-Aldrich, Merck.

### 4.3. RNA Extraction and Real-Time Reverse Transcriptase-PCR

Total cellular RNA was extracted using a ReliaPrep RNA Cell Miniprep System (Promega, Madison, WI, USA; Z6011), and first-strand DNA was synthesized using a cDNA Reverse Transcription Kit (Applied Biosystems, Thermo Fisher Scientific Inc., Waltham, MA, USA; 4368814), following the manufacturer’s instructions. Real-time quantitative reverse transcriptase PCR (RT-qPCR) was performed on a CFX Connect Real-Time System (Bio-Rad, Hercules, CA, USA) using PowerTrack SYBR Green Master Mix (Applied Biosystems, Thermo Fisher Scientific Inc.; A46109) according to the standard protocol. The expression of the target mRNA was quantified relative to that of the endogenous control mRNA, GAPDH. Untreated controls were used as a reference. Sequences of the primers are provided in Appendix A.

### 4.4. siRNA Transfection

For Ankrd1 silencing, 786-O and 769-P cells were transfected with three siRNAs (5 nM) using Lipofectamine RNAiMAX (Invitrogen, Thermo Fisher Scientific Inc., Waltham, MA, USA; 13778), according to the manufacturer’s recommendations. Gene-specific and nonspecific control (DS NC-1) siRNAs were purchased from Integrated DNA Technologies, Inc., Coralville, IA, USA, and their targeting sequences are provided in Appendix A.

### 4.5. Overexpression of Plasmid DNA

A-498 and 786-O cells, cultured in 6-well plates, were transfected with empty or 3xFlag-Ankrd1-pcDNA3.1 vectors (2.5 µg DNA/well) using the Lipofectamine 3000 Transfection Kit (Invitrogen, Thermo Fisher Scientific Inc.; L3000). The plasmid was kindly provided by Dr. Francesco Acquati (University of Insubria, Vares, Italy) [43].

### 4.6. Cell Proliferation Assay

Cells were seeded in 96-well plates at two seeding densities, 3 × 10^3^ and 0.8 × 10^3^ per well. The next day, the cells were transfected with siRNA against Ankrd1 or DS-NC-1. Cell proliferation assay was performed after 72 h of siRNA transfection. Cell viability was estimated as %OD value using the CellTiter 96 Aqueous One Solution Cell Proliferation Assay (Promega, G3580), as described previously [42].

### 4.7. Immunoblotting

Immunoblotting analysis was performed using the Western BLoT Hyper HRP Substrate (TaKaRa Bio Inc., Shiga, Japan; T7103A) and Western BLoT Ultra HRP Substrate (TaKaRa Bio Inc.; T7104A), as described previously [42]. The details of the primary antibodies are provided in Appendix A.

### 4.8. Cell Cycle Analysis

Cell cycle analysis was performed using flow cytometry, as described previously [42]. 786-O cells transfected with siRNA against Ankrd1 or DS NC-1 were incubated with phosphate-buffered saline (PBS) containing 50 µg/mL propidium iodide (Sigma-Aldrich, Merck; P4864) and 20 µg/mL Ribonuclease A (Sigma-Aldrich, Merck; P4642) at 37 °C for 30 min. The stained cells were analyzed using a BD FACSMelodyTM system (BD Life Sciences-Biosciences, Franklin Lakes, NJ, USA).

### 4.9. Immunoprecipitation Assay

786-O cells transfected with empty or 3xFlag-Ankrd1-pcDNA3.1 vector were lysed in ULTRARIPA kit for Lipid Raft A buffer (BioDynamics Laboratory Inc., Tokyo, Japan; F012). We used the lysate of 786-O cells transfected with the empty pcDNA3.1 vector as a control. For immunoprecipitation, ANTI-FLAG M2 Affinity Gel (Millipore, Merck, Darmstadt, Germany; A2220) was used in accordance with the manufacturer’s protocol. The lysate (3 mg total protein in 600 µL volume) was added to 40 µL ANTI-FLAG M2 Affinity Gel and incubated overnight on a rotator at 4 °C. The next day, the gel was washed three times with 1000 µL Tris-buffered saline, and immunocomplexes were eluted with 3 × FLAG peptide (Sigma-Aldrich, Merck; F4799). The whole-cell lysate was loaded as a control input. The samples were analyzed using immunoblotting, as described in the Immunoblotting section.

### 4.10. Wound-Healing Assay

Wound-healing assay was performed as described previously [44]. Before seeding in six-well plates, 786-O and 769-P cells were transfected with siRNA, and A-498 was transfected with Flag-Ankrd1 or a control plasmid. After the cells reached confluence, the surface of the wells was scratched in a straight line with a pipette tip. The wound area was measured using a BZ-X700 microscope (Keyence Co., Osaka, Japan) and the ImageJ software version 1.53 at 0 and at 8 (786-O), 12 (769-P), and 17 (A-498) hours. Wound closure % was determined as the percentage reduction in the wound area compared with the initial wound area.

### 4.11. Invasion Assay

Cell invasion assay was performed as described previously [44]. 786-O and 769-P cells were transfected with siRNA, and A-498 was transfected with Flag-Ankrd1 or control plasmid. Cells in serum-free RPMI medium (75 × 10^3^ per well) were seeded on the top of a Matrigel-coated insert (Corning International, Tokyo, Japan; 354480) or in a non-coated control chamber (Corning International; 353097). After 24 h of incubation, the cells at the bottom of the inserts were stained with crystal violet and imaged using an BX43 microscope (Olympus Corporation, Tokyo, Japan). The invading cells were counted under 200× magnification in five representative fields using the ImageJ software. The % invasion was defined as the percentage of cells that crossed the membrane in the presence of the Matrigel layer versus cells that crossed it in the absence of the Matrigel layer.

### 4.12. Measurement of Cell Size

786-O cells transfected with siRNA against Ankrd1 or DS-NC-1 were analyzed for cell area. After 48 h of incubation, the viable cells were imaged using a Primovert microscope (Carl Zeiss, Oberkochen, Germany) and morphologically examined. The ImageJ software was used to draw the outline of a colony of cells and to measure the area. The area was then divided by the number of cells.

### 4.13. Immunofluorescence Analysis

For confocal microscopy, 786-O cells were seeded on collagen-coated glass-bottom dishes. The cells were stimulated by incubating with the PKC activator, TPA (150 nM, 20 min). The cells were fixed with 4% paraformaldehyde for 15 min and permeabilized using 0.1% Triton X-100 for 10 min. After blocking with 1% BSA for 30 min, the cells were incubated overnight at 4 °C with primary antibodies (Appendix A) and subsequently with Alexa Fluor 488 or 555-labeled secondary antibodies (Cell Signaling Technology Japan). Actin filaments were stained with Alexa Fluor 555 Phalloidin Molecular Probes (Cell Signaling Technology Japan; 4409), and the nuclei were stained with Hoechst 33342 (Dojindo, Kumamoto, Japan; 346-07951). The fluorescence was visualized using confocal laser scanning microscopy (Zen; Carl Zeiss).

### 4.14. Proximity Ligation Assay

786-O cells cultured on collagen-coated glass-bottom dishes were fixed and permeabilized as described for the immunofluorescence analysis. The cells were processed using Duolink in Situ PLA Probe Anti-Mouse PLUS (Sigma-Aldrich, Merck; DUO92001), Probe Anti-Rabbit MINUS (Sigma-Aldrich, Merck; DUO92005), and Detection Reagents FarRed (Sigma-Aldrich, Merck; DUO92013), following the manufacturer’s protocol. Rabbit anti-Ankrd1 and mouse anti-talin-1 antibodies were used for the assay (Appendix A). We used the sample incubated without the anti-talin-1 antibody as a negative control.

### 4.15. Immunohistochemistry

RCC samples, along with the clinicopathological information, were obtained from the patients who had undergone surgeries for both primary and metastatic lesions of RCC at Yamagata University Hospital between 2003 and 2021. Twenty-six metastatic lesions and 20 corresponding primary lesions were collected for IHC, which was performed using standard procedures [45]. The epitopes were activated by autoclaving in 10 mM citrate buffer (pH 6.0) at 120 °C for 20 min. Details of the primary antibodies used are provided in Appendix A. For negative control, the primary antibody was replaced with anti-rabbit isotype control IgG (Cell Signaling Technology Japan; 3900; RRID:AB_1550038). Human cardiac muscle tissue was used as a positive tissue control.

As previously described [46], the staining intensity was scored as follows: 0: negative; 1: weak; 2: moderate; 3: strong. The percentage of positive cells was scored as follows: 0: 0%; 1: 1–10%; 2: 11–50%; 3: 51–80%; 4: >80%. The immunostaining intensity and average percentage of positive cells were evaluated for observations made in five independent high magnification fields. The IHC score (0–12) was obtained by multiplying the scores for the staining intensity and percentage of positive cells.

### 4.16. Statistical Analysis

For bar charts, continuous variables are presented as means ± standard deviation (SD). They were statistically analyzed using the *t*-test, analysis of variance (ANOVA), and, if necessary, a post hoc Bonferroni test for multiple comparisons. The Mann–Whitney *U* test and Wilcoxon signed-rank test were used to compare the distribution of the IHC scores. A *p*-value < 0.05 was considered to indicate statistical significance. All analyses were performed using the EZR software version 1.54 (Saitama Medical Center, Jichi Medical University, Saitama, Japan).

## 5. Conclusions

We show that Ankrd1 promotes the motility and metastases of RCC cells. Ankrd1 is essential for the formation of lamellipodia via its interaction with talin-1 and F-actin. Therefore, blocking the binding of Ankrd1 and talin-1 might be a new therapeutic strategy.

## Figures and Tables

**Figure 1 ijms-26-04232-f001:**
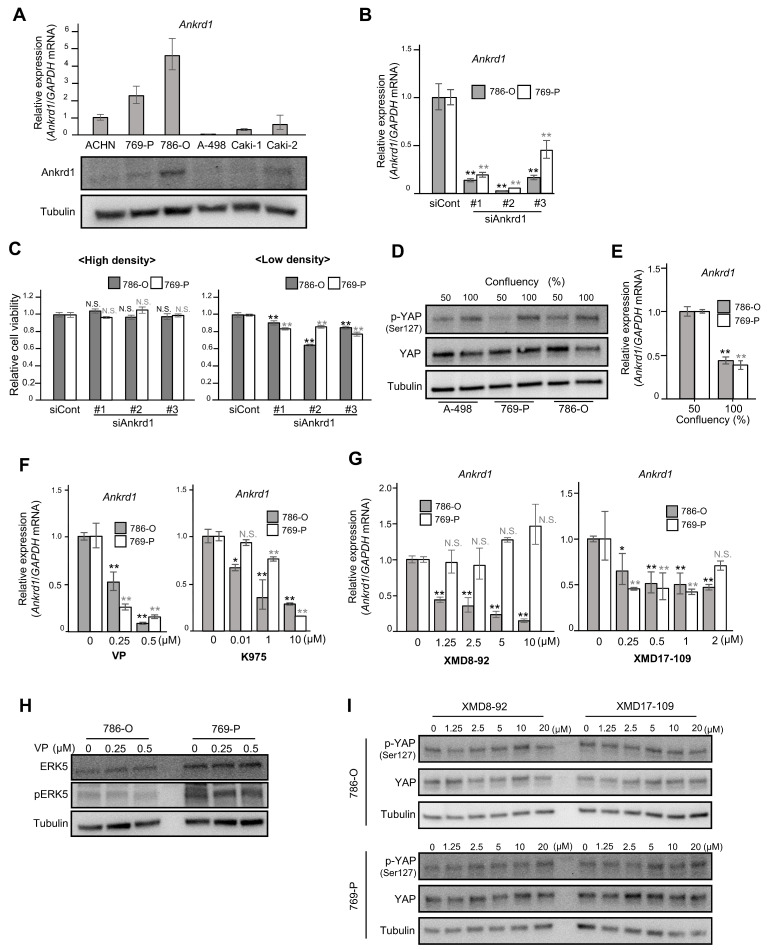
Ankrd1 expression is regulated by YAP, and it affects the proliferation of 786-O cells at low confluency. (**A**) Relative expression of Ankrd1 in renal cell carcinoma (RCC) cell lines measured using RT-qPCR and Western blotting analyses. (**B**) Knockdown of Ankrd1 in 786-O and 769-P cells using siRNAs, as confirmed via RT-qPCR. (**C**) Relative viability of 786-O cells transfected with siAnkrd1 measured using the MTS assay. The cells were seeded into 96-well plates at a density of 3 × 10^3^ (left) or 0.8 × 10^3^ (right) cells/well. Ankrd1 knockdown decreased the cell viability only at low cell density. (**D**) Phosphorylation levels of YAP^ser127^ at different confluences. (**E**) Ankrd1 expression at different confluences. (**F**) Treatment with YAP inhibitors (48 h) decreased Ankrd1 expression in 786-O and 769-P cell lines. VP, verteporfin. (**G**) Treatment with ERK5 inhibitors (48 h) decreased Ankrd1 expression in 786-O and 769-P cell lines. (**H**) Expression of ERK5 and p-ERK5 in 786-O and 769-P cell lines, 48 h after treatment with verteporfin. (**I**) Phosphorylation levels of YAP in 786-O and 769-P cell lines, 30 min after treatment with ERK5 inhibitors. Values represent means ± SD. ** *p* < 0.01; * *p* < 0.05; N.S., Not significant compared with control.

**Figure 2 ijms-26-04232-f002:**
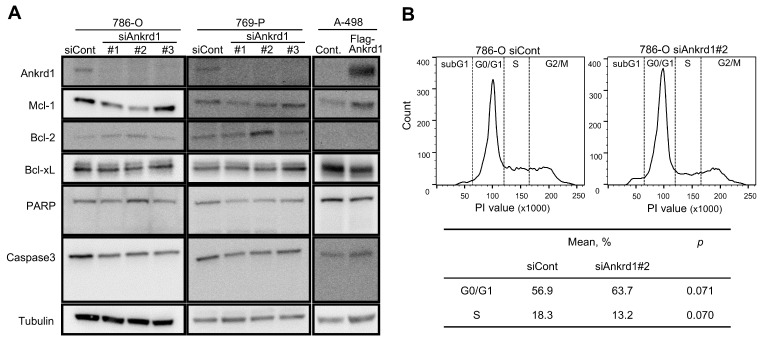
Effect of Ankrd1 on apoptosis and cell cycle. (**A**) Expression of apoptosis-related proteins in 786-O and 769-P cells transfected with siAnkrd1 or siCont, and in A-498 cells transfected with Flag-Ankrd1 or control vector. (**B**) Cell cycle distribution, 72 h after the transfection with siAnkrd1 or siCont.

**Figure 3 ijms-26-04232-f003:**
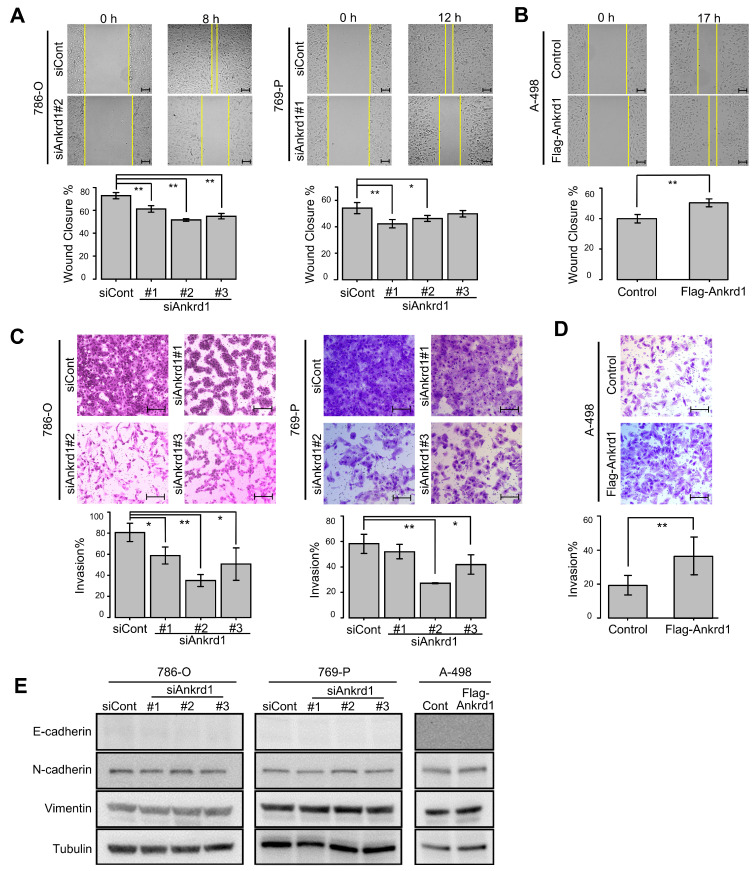
Knockdown of Ankrd1 suppressed cell migration and invasion. (**A**) Wound-healing assay in 786-O and 769-P cells transfected with siAnkrd1 or siCont. Scale bar, 100 µm. (**B**) Wound-healing assay in A-498 cells transfected with Flag-Ankrd1 or control vector. Cell migration was accelerated by overexpression of Ankrd1. Scale bar, 100 µm. (**C**) Invasion assay for 786-O and 769-P cells transfected with siAnkrd1. Scale bar, 200 µm. (**D**) Invasion assay for A-498 cells transfected with Flag-Ankrd1 or control vector. Cell invasion was accelerated by overexpression of Ankrd1. Scale bar, 200 µm. (**E**) Expression of epithelial–mesenchymal transition-related proteins in 786-O and 769-P cells transfected with siAnkrd1 or siCont and in A-498 cells transfected with Flag-Ankrd1 or control vector. Values represent means ± SD. ** *p* < 0.01; * *p* < 0.05.

**Figure 4 ijms-26-04232-f004:**
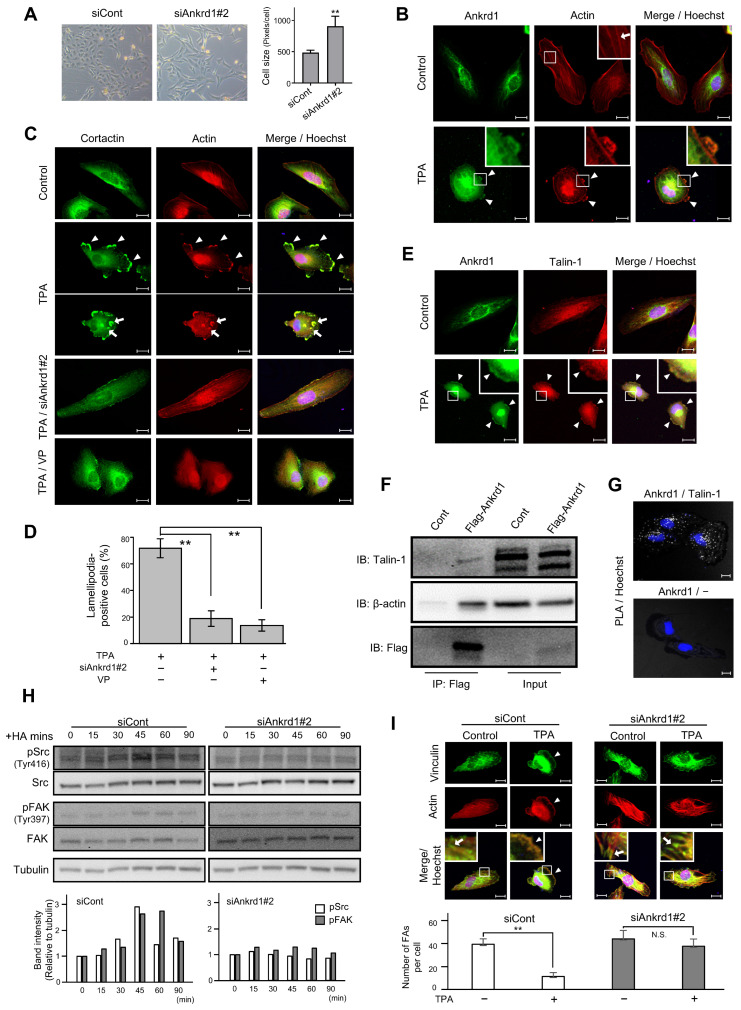
Immunofluorescence analysis of Ankrd1 and related proteins in 786-O cells. (**A**) Change in the shape and size of 786-O cells after siAnkrd1 transfection observed using light microscopy. (**B**) Subcellular localization of Ankrd1 and actin. 12-O-Tetradecanoylphorbol 13-acetate (TPA) exposure (150 nM, 20 min) induced lamellipodia (arrowhead). The arrow shows stress fiber. Hoechst staining depicts the nucleus. (**C**) Localization of cortactin in TPA-induced lamellipodia (arrowhead) and invadopodia (arrow). siAnkrd1 and verteporfin treatment blocked TPA-induced lamellipodia. (**D**) Ratio of cells with lamellipodia. Values represent means ± SE. ** *p* < 0.01. (**E**) Colocalization of Ankrd1 and talin-1 in TPA-induced lamellipodia (arrowhead). (**F**) Coimmunoprecipitation of Flag-Ankrd1 with talin-1 and β-actin in 786-O cells. (**G**) Proximity ligation assay (PLA) showing that Ankrd1 and talin-1 are in close proximity. (**H**) Changes in the phosphorylation levels of Src and FAK over time after HA treatment. (**I**) Localization of vinculin after TPA exposure. siAnkrd1 reduced the localization of vinculin to TPA-induced lamellipodia (arrowhead) from focal adhesion (FA) (arrow). The number of FAs per cell was evaluated. Values represent means ± SD. ** *p* < 0.01; N.S., Not significant compared with control. Scale bar, 20 µm.

**Figure 5 ijms-26-04232-f005:**
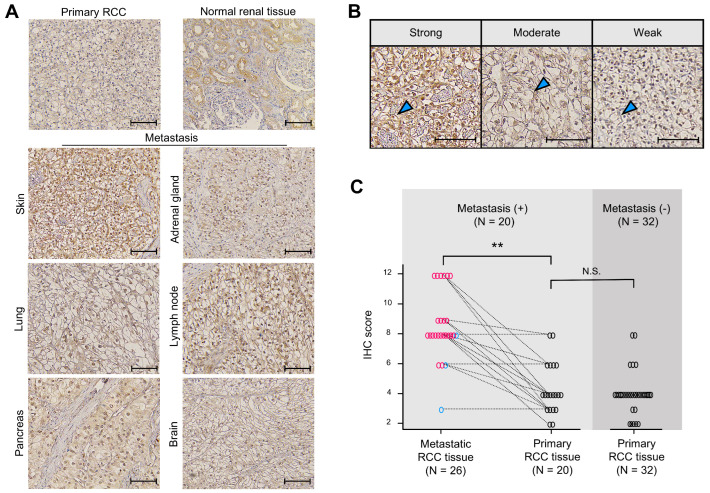
Ankrd1 expression is increased in metastatic renal cell carcinoma (RCC) tissues. (**A**) Representative immunohistochemistry (IHC) images of primary (normal renal and RCC tissues) and metastatic RCC tissues. Scale bar, 100 µm. (**B**) Representative images of sections showing different signal intensities of stained cell membranes (arrowhead). Scale bar, 100 µm. (**C**) IHC scores for stained cell membrane in RCC tissue samples. IHC scores were significantly higher in metastatic RCC tissues than in primary RCC tissues. The dashed lines indicate paired samples of primary and metastatic lesions from the same case. ** *p* < 0.01; N.S., Not significant.

**Table 1 ijms-26-04232-t001:** Patient demographics.

Clinical Variable	Metastasis (+) (*N* = 20)	Metastasis (−) (*N* = 32)	*p*
Age at surgery of primary lesion (year), mean (range)	60.7 (44–76)	68.4 (34–91)	
Sex, *n* (%)			1
Men	17 (85.0)	27 (84.4)	
Women	3 (15.0)	5 (15.6)	
Subtype, *n* (%)			1
Clear	20 (100)	32 (100)	
pT stage, *n* (%)			<0.01
1a	2 (10.0)	26 (81.3)	
1b	7 (35.0)	5 (15.6)	
2a	1 (5.0)	1 (3.1)	
2b	2 (10.0)	0 (0)	
3a	8 (40.0)	0 (0)	
Fuhrman grade, *n* (%)			0.658
1	2 (10.0)	8 (25.0)	
2	12 (60.0)	15 (46.9)	
3	5 (25.0)	7 (21.9)	
Grade, *n* (%)	1(5.0)	2 (6.2)	
Metastasis, *n* (%)			
Lung	11 (42.3)		
Adrenal gland	4 (15.4)		
Skin	3 (11.5)		
Pancreas	3 (11.5)		
Lymph node	3 (11.5)		
Brain	1 (3.8)		
Vein	1 (3.8)		

**Table 2 ijms-26-04232-t002:** Correlation between Ankrd1 expression and pathologic features in clear cell renal cell carcinoma.

Pathological Variables	IHC Score (Mean)	*p*
Fuhrman grade		0.414
1, 2 (*n* = 37)	4.35	
3, 4 (*n* = 15)	3.80	
pT stage		0.493
1a, 2a (*n* = 42)	4.12	
2b, 3a (*n* = 10)	4.50	

## Data Availability

The datasets generated during the current study are available from the corresponding author on reasonable request.

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
