# Peer review of "Ankrd1 Promotes Lamellipodia Formation and Cell Motility via Interaction with Talin-1 in Clear Cell Renal Cell Carcinoma"

_ijms, 2025, doi:10.3390/ijms26094232_

Round 1

Reviewer 1 Report

Comments and Suggestions for Authors

Summary

The authors are interested in renal cancer (Clear Cell Renal Cell Carcinoma, ccRCC) and the role of ANKRD1. The rationale for trying to link ANKRD1 to ccRCC is not well made but the experiments are conducted in a logical manner and the need to learn more about ccRCC and the role of ANKRD1 in general is explained well.

The study begins with an informatic analysis of public datasets highlighting a role for ANKRD1 in ccRCC, and subsequently confirms the expression of ANKRD1 in a panel of relevant cell lines and metastatic ccRCC. The authors show through silencing and overexpression of ANKRD1 that the protein has a role in cell motility in vitro. Mechanistically ANKRD1 localised with actin in lamellipodia and colocalised with talin, a key protein involved in integrin activation and signalling.

In this reviewer’s opinion this work is performed and presented to an appropriate standard for publication and will be of interest to a wide readership i.e. those studying cancer (ccRCC in particular), cell adhesion and cytoskeletal function. I am supportive of publication but have raised several issues below that should be considered before doing so.

Main issues:

  1. The results described in the abstract do not reflect the order of data as presented in the results section of the manuscript e.g. the expression of ANKRD1 in tumours is in the final figure 5 but listed 2nd in the abstract. This was confusing for this reviewer.
  2. Fig 3. Migration images not clear. The authors have provided guide lines (yellow) but it is currently very hard to see phase contrast images of the cells themselves.
  3. Fig 3. Migration. I like use of multiple siRNA but more conclusive evidence would be to add in rescue experiments i.e. can FLAG-ANKRD1 expression rescue effects of KD?
  4. The authors note in Fig 4 that ANKRD1 KD resulted in larger cells. Were these changes quantified or also observed in the flow cytometry data obtained for cell cycle analysis? (fig 2)
  5. Fig 4. IF images in panels could be larger to assist the reader see subcellular localisations? For example, the authors claim that actin stress fibres are visible but this is not clear in the images presented. Also, the talin localisation is not clear. This reviewer cannot see the expected punctate focal adhesion type staining. Could the figure panels be split over 2 figures to help?
  6. Fig 4F and G. Important controls are either not detailed or provided to support the specificity of the Talin / ANKRD1 pulldowns / proximity.
    1. What is ‘Cont’ in panel D? Ideally this should be cells expressing a FLAG-tagged protein that is not ANKRD1.
    2. For panel 4G please detail controls used for the PLA approach. Could this be performed in siANKRD1 KD cells too?
    3. Did the authors blot for any proteins that were not detected in the FLAG pull downs in ANKRD1 expressing cells? The data for the specific association with talin and actin would be more convincing if other cytoskeletal or membrane proteins were not detected in this pull down or by PLA.
    4. Have the authors tried an co-IP with endogenous proteins i.e. not over-expressed fusion protein or in a reciprocal way i.e. IP talin and blot for FLAG-ANKRD1?
  7. Fig 4C/D the rationale and interpretation for use of the YPA inhibitor, verteporfin, is not clear.
  8. Fig 4 G, H and I. Could the authors provide quantification of the effects observed in these panels / experiments
  9. Fig 5. Could the authors provide additional labels to indicate which structures are considered positive (High/Low) or negative for ANKRD1 expression in the IHC images
  10. Page 10 line 280. Please check / remove comment M1 about section 2.2
  11. Discussion and Abstract: that authors claim that the data presented show an interaction between ANKRD1 and talin which is not the case. To make this claim data would be needed to show that purified proteins interact. The data suggests they are in close proximity by IF/PLA and can be isolated in a complex (co-IP). Please modify the language used througouht the manuscript.
  12. Title page. Affiliations are not clear. Please provide full institutional addresses and official emails.

Reviewer 2 Report

Comments and Suggestions for Authors

I find the submitted manuscript interesting and worthwhile. Nevertheless, I believe that it could be significantly improved by clarifying some points that I find hard to understand and that make the manuscript unclear. My suggestion is that the enlisted points, which I am enclosing, would highlight the impact of the manuscript.

  1. In the results section 2.1 heading, the word “Subsection” is unnecessary. Figure 1A should be considered as a one figure because the text in the results are very specific. The reason is that the results included in Figure 1B-J show unrelated results regarding the heading of the section.
  2. Figure 1E, G, I and J, results include YAP and ERK5 detection but the authors do not mention anything about these proteins in the introduction section. Why are these proteins relevant? How do they relate with Ankrd?
  3. Figure 1 extensive footnote heading is too large and confusing; it does not describe the results. It mentions that YAP increases 786-O cell proliferation but these results are not shown in the figure.
  4. In the results section 2.3, the authors conclude that “Because Ankrd1 decreased the viability of 786-O and 769-P cells only 157 at high cell density, apparently Ankrd1 has a small impact on the viability of RCC cells”. Where are the results to support this conclusion?
  5. Figure 2 footnote legend is wrong. The authors mention in the text that changes in the proportion of G0/G1 and S phase are nonsignificant, therefore I do not understand why does the figure footnote conclude that Ankrd promotes the proliferation of RCC cells?
  6. Something in figure 3 statistical analysis is probably wrong. According with the standard deviation shown in the histograms, there are not differences in the migration or invasion assays. It is mandatory to verify the analysis of the data.
  7. If EMT-related proteins were not affected by Ankrd, which other markers can or could the authors test in order to verify its participation on the metastatic process?
  8. Figure 4 shows Cortactin, Talin, and Vinculin distribution when 786-O cells are incubated with TPA. However, Section 2.5 says, “Interaction of Ankrd1 and talin-1 is Involved in the Formation of Lamellipodia”. Where is the result that represents that conclusion? The image that shows the PLA assay is not clear enough.

Round 2

Reviewer 1 Report

Comments and Suggestions for Authors

The authors have addressed the comments I raised. It appears that they may have been given only a short time frame to do this so it is not reasonable to expect any additional experiments to have been performed.

Having said that I think there are 2 comments that could be addressed better by the addition of more detail of experiments performed with negative results.

For example the responses 6-3 and 6-4 contain very useful information that is worth sharing. It will strengthen the findings and not be a weakness to provide these details.

Reviewer 2 Report

Comments and Suggestions for Authors

I agree with the corrections that the authors made, now it is clearer.